# Pin-Pointing the Key Hubs in the IFN-γ Pathway Responding to SARS-CoV-2 Infection

**DOI:** 10.3390/v14102180

**Published:** 2022-09-30

**Authors:** Ayelen Toro, Sofia Lage-Vickers, Juan Bizzotto, Felipe Vilicich, Agustina Sabater, Gaston Pascual, Sabrina Ledesma-Bazan, Pablo Sanchis, Maria Sol Ruiz, Ana Paula Arevalo, Jorge L. Porfido, Mercedes Abbate, Rocio Seniuk, Estefania Labanca, Nicolas Anselmino, Nora M. Navone, Daniel F. Alonso, Elba Vazquez, Martina Crispo, Javier Cotignola, Geraldine Gueron

**Affiliations:** 1Laboratorio de Inflamación y Cáncer, Departamento de Química Biológica, Facultad de Ciencias Exactas y Naturales, Universidad de Buenos Aires, Buenos Aires C1428EGA, Argentina; 2Instituto de Química Biológica de la Facultad de Ciencias Exactas y Naturales (IQUIBICEN), CONICET—Universidad de Buenos Aires, Buenos Aires C1428EGA, Argentina; 3Instituto de Tecnología (INTEC), Universidad Argentina de la Empresa (UADE), Buenos Aires C1073AAO, Argentina; 4Laboratory Animals Biotechnology Unit, Institut Pasteur de Montevideo, Montevideo 11400, Uruguay; 5Department of Genitourinary Medical Oncology and The David H. Koch Center for Applied Research of Genitourinary Cancers, The University of Texas MD Anderson Cancer Center, Houston, TX 77030, USA; 6Centro de Oncología Molecular y Traslacional y Plataforma de Servicios Biotecnológicos, Departamento de Ciencia y Tecnología, Universidad Nacional de Quilmes, Bernal B1876BXD, Argentina

**Keywords:** IFN-γ, ISGs, COVID-19

## Abstract

Interferon gamma (IFN-γ) may be potential adjuvant immunotherapy for COVID-19 patients. In this work, we assessed gene expression profiles associated with the IFN-γ pathway in response to SARS-CoV-2 infection. Employing a case-control study from SARS-CoV-2-positive and -negative patients, we identified IFN-γ-associated pathways to be enriched in positive patients. Bioinformatics analyses showed upregulation of *MAP2K6*, *CBL*, *RUNX3*, *STAT1*, and *JAK2* in COVID-19-positive vs. -negative patients. A positive correlation was observed between *STAT1*/*JAK2*, which varied alongside the patient’s viral load. Expression of *MX1*, *MX2*, *ISG15*, and *OAS1* (four well-known IFN-stimulated genes (ISGs)) displayed upregulation in COVID-19-positive vs. -negative patients. Integrative analyses showcased higher levels of ISGs, which were associated with increased viral load and *STAT1*/*JAK2* expression. Confirmation of ISGs up-regulation was performed in vitro using the A549 lung cell line treated with Poly (I:C), a synthetic analog of viral double-stranded RNA; and in different pulmonary human cell lines and ferret tracheal biopsies infected with SARS-CoV-2. A pre-clinical murine model of Coronavirus infection confirmed findings displaying increased ISGs in the liver and lungs from infected mice. Altogether, these results demonstrate the role of IFN-γ and ISGs in response to SARS-CoV-2 infection, highlighting alternative druggable targets that can boost the host response.

## 1. Introduction

Coronaviruses are a family of viruses that can cause several illnesses, such as the common cold, severe acute respiratory syndrome (SARS), and Middle East respiratory syndrome (MERS). Since 2002, there have been three different outbreaks caused by coronaviruses: in 2002, there was a SARS-CoV epidemic; in 2012, there was a MERS-CoV outbreak; and in 2020 the novel Coronavirus SARS-CoV-2 became a pandemic, being the third Coronavirus outbreak to emerge in the human population [1].

Interferons (IFNs) are the most important innate antiviral cytokines. Type I and type III IFNs (IFN-I and IFN-III, respectively) are expressed by most cell types and have the ability to induce an antiviral response within infected and surrounding cells [2]. Regarding IFN-I’s role in COVID-19, it was demonstrated that numerous SARS-CoV-2 proteins have the ability to inhibit IFN-I production and/or IFN-I responses [3], producing resistance to this antiviral mechanism [4]. Although viruses display strategies to evade the IFN antiviral activity, there are host factors that affect the severity of the disease. For example, the generation of auto-antibodies that target IFN-I has the potential to affect the course of SARS-CoV-2 infection, suggesting that IFN deficiency may result in severe COVID-19 [5,6]. Further, an enrichment in loss-of-function variants in IFN-I pathway-related genes was reported in severe COVID-19 cases [7]. In addition, Hadjadj et al. observed that peripheral blood immune cells from severe and critical COVID-19 patients had diminished IFN-I production and enhanced proinflammatory cytokines [8]. Hence, severe COVID-19 patients might be potentially relieved from the IFN-I deficiency through IFN administration [8]. Concerning the IFN-III association with COVID-19, reduced protein levels of IFN-λ were also correlated with higher disease severity [9]. Regarding IFN-II, there is scarce evidence linking it to COVID-19. Interestingly, serum levels of IFN-γ, the type II IFN (IFN-II), were significantly increased when comparing symptomatic vs. asymptomatic COVID-19 groups [10]. IFN-γ is predominantly produced by immune cells. However, the IFN-γ receptor (IFNGR), which is composed of two subunits (IFNGR1 and IFNGR2), is ubiquitously expressed. Therefore, all cell types are capable of responding to IFN-γ signals. Binding of IFN-γ to the IFNGR1/2 results in the activation of the Janus Kinases 1 and 2 (JAK1 and JAK2), leading to the phosphorylation and homodimerization of the Signal Transducer and Activator of Transcription 1 (STAT1) [11]. STAT1 dimers, also known as the gamma interferon-activated factor (GAF), translocate to the nucleus and promote gene expression by binding to the gamma interferon-activated site (GAS) of IFN-stimulated genes (ISGs) [12]. In addition, after IFNGR stimulation, STAT1 is able to interact with other proteins such as STAT2 and Interferon Related Factor 9 (IRF9), which, in turn, regulate IFN-γ-responsive genes by binding to the interferon-stimulated response elements (ISRE) [12]. Consequently, IFN exposure triggers the induction of more than 300 ISGs, such as the Myxovirus resistance genes 1 and 2 (*MX1* and *MX2*), 2′,5′-oligoadenylate synthetase 1 (*OAS1*), and Interferon-stimulated gene 15 (*ISG15*), that collectively promote an antiviral response [13,14].

There are several studies that highlight IFNs as key players in COVID-19 and propose that a dysregulated IFN signaling pathway is associated with a poor prognosis [5,7,9]. SARS-CoV-2 has evolved several strategies to counteract not only IFN production but also IFN signaling. In fact, 16 viral proteins target the host’s IFN pathway at several levels to escape IFN-mediated restriction [15]. Although much attention has been placed on the SARS-CoV-2 mediated evasion of the IFN response, not much attention has been given to the role of IFN-γ-associated genes in COVID-19.

In this work, we undertook an extensive bioinformatics analysis in human patients, positive or negative for SARS-CoV-2, to evaluate gene expression profiles associated with the IFN pathways, focusing on the canonical and non-canonical IFN-γ axes. Validation of results was performed in vitro and in vivo in ferrets and in a murine Coronavirus model of infection. Our results evidence the involvement of the IFN-γ signaling pathway in COVID-19, pointing out a potential mechanism involved in SARS-CoV-2 infection, highlighting alternative druggable targets that may boost the host response against SARS-CoV-2.

## 2. Materials and Methods

### 2.1. Transcriptome Datasets Selection

To identify potential relevant studies with transcriptome data related to SARS-CoV-2 infection and COVID-19 patients, we browsed the following databases: Gene Expression Omnibus (GEO) repository [16], Genotype-Tissue Expression (GTEx) project [17], Genome Sequence Archive (GSA) on the National Genomics Data Center [18], PubMed [19], and Google Scholar [20]. We used the following keywords and expressions: [(COVID) OR (COVID-19) OR (CORONAVIRUS) OR (SARS-CoV-2) OR (2019-nCoV)] AND [(transcriptomics) OR (RNA-seq) OR (microarray) OR (expression) OR (transcriptome)].

All potentially relevant datasets were further evaluated in detail by 2–3 authors. The eligibility criteria included: (i) publicly available transcriptome data (gene expression microarray, RNA-seq); (ii) detailed sample information; (iii) detailed protocol information. We selected the following datasets that complied with these criteria:

GSE147507 [21]: RNA-seq data from (i) human cell lines derived from primary bronchial/tracheal epithelial cells (NHBE), lung carcinoma (hACE2 A549), and lung adenocarcinoma (Calu-3) infected with SARS-CoV-2 (MOIs: 2, 0.2 and 2, respectively) or mock-PBS; and (ii) tracheal samples from ferrets intranasally infected with 5 × 10^4^ PFU of SARS-CoV-2 or mock-PBS.

GSE152075 [22]: RNA-seq data from 430 SARS-CoV-2-positive (COVID-19) and 54 negative patients (non-COVID-19) diagnosed by RT-qPCR. Clinico-pathological information included age, sex, and viral load. The cycle threshold (Ct) by RT-qPCR for the N1 viral gene at the time of diagnosis was used to determine the viral load. The interpretation for the viral load was: the higher the viral load, the lower the Ct.

GSE32138 [23]: Whole Human Genome Microarray from human airway epithelial cells from non-cystic fibrosis patients resected at lung transplantation. Cells were infected with Influenza A Virus (IAV, 2 × 10^5^ PFU) or Respiratory Syncytial Virus (RSV, 1×10^6^ PFU) or mock and harvested 24 h or 48 h post infection, respectively. 

GSE100504 [24]: Whole Human Genome Microarray from human airway epithelium cultures. Cells were infected with wild-type MERS-CoV (MOI 5) or mock for 48 h. 

GSE47963 [25]: Whole Human Genome Microarray from human airway epithelium cultures. Cells were infected with wild-type SARS-CoV (MOI 2) or mock for 48 h. 

### 2.2. Bioinformatics Analyses

#### 2.2.1. RNA-Seq Analysis

For the GSE147507 dataset, we downloaded the raw RNA-seq data and checked the quality of reads by FastQC software. Sequences were aligned to the reference genome GRCh38 using STAR aligner [26], normalized by transcript length and GC content using the hg38 genome annotation with metaseqR package in R [27]. Additionally, within and between lanes, normalization was performed using the EDAseq package in R [28]. For the GSE152075 dataset, we proceeded as previously described in Bizzotto et al. [29]. 

The comparisons were plotted using pheatmap [30], ggplot2 [31], and ggpubr [32] packages in R. Student’s *t*-test was performed to determine statistical differences.

#### 2.2.2. Microarray Analysis

For the GSE32138, GSE100504 and GSE47963 datasets, we downloaded the normalized gene expression data from all microarray datasets. The comparisons were plotted using pheatmap [30], ggplot2 [31], and ggpubr [32] packages in R. Student’s *t*-test was performed to determine statistical differences.

### 2.3. Gene Correlation

Pairwise gene correlation between the IFN-γ-associated genes that were differentially expressed in COVID-19-positive vs. -negative patients was analyzed with the ggcorrplot package in R. Correlation coefficients were classified as weak (|r| ≤ 0.33), intermediate (0.33 < |r| < 0.66), and strong (|r| ≥ 0.66). Statistical significance was set at *p* < 0.05.

### 2.4. Profiling of Immune Cell Type Abundance from Bulk Gene Expression Data Using CIBERSORT

CIBERSORT web tool [33] was used to estimate the abundances of immune cell types based on the LM22 signature on normalized bulk RNA-seq gene expression data. Results were displayed as heatmaps by non-supervised clustering on rows and columns using pheatmap function in R [30]. 

### 2.5. Gene Set Variation Analysis (GSVA)

Assessment of gene set enrichment was performed on normalized RNA-seq gene expression data using Gene Set Variation Analysis (GSVA) tool in R [34]. GSVA uses a non-parametric, unsupervised method that allows the estimation of the relative enrichment of pathways across samples. “C2:KEGG”, “C2:REACTOME”, “C5” and “C7” geneset collections were downloaded using msigdbr [35] package in R. Results were filtered, and only genesets that include “interferon”, “IFN”, and “gamma” were used for subsequent analysis and visualization. 

### 2.6. Cell Culture Conditions

Human lung carcinoma A549 (ATCC^®^ CCL-185™) cells were grown in Dulbecco’s modified Eagle’s medium (DMEM, Gibco, Rockville, MD, USA) plus 10% fetal bovine serum (FBS) (Internegocios, Buenos Aires, Argentina), penicillin 100 U/ml, streptomycin 100 µg/mL, amphotericin 0.5 µg/mL, 2 mM glutamine, and 80 μg/mL gentamycin in monolayer culture, at 37 °C in a humidified atmosphere of 5% CO_2_. Cells were harvested using a trypsin/EDTA solution (Gibco) diluted in PBS and routinely tested for mycoplasma.

### 2.7. Poly (I:C) Treatment

Poly (I:C) intracellular administration was performed by transfection with Lipofectamine LTX (Invitrogen, Carlsbad, CA, USA) according to the manufacturer’s instructions at a final concentration of Poly (I:C) of 10 µg/mL. Briefly, A549 cells were plated in 6-well flat bottom plates at a density of 2 × 10^5^ cells in complete DMEM, allowed to attach overnight, and then transfected with Poly (I:C) or distilled water as mock in DMEM without FBS and antibiotics. After 6 h, the media were replaced, and 24 h later, total cell lysates and RNA isolation was carried out [36].

### 2.8. In Vivo Experiments

BALB/cJ female mice (n = 10; 8–10 weeks old) were bred at the animal facility of the Laboratory Animals Biotechnology Unit of Institut Pasteur de Montevideo under specific pathogen-free conditions in individually ventilated racks (IVC, 1285L, Tecniplast, Milan, Italy). The housing environmental conditions during the experiment were as follows: 20 ± 1 °C temperature, 30–70% relative humidity, negative pressure (biocontainment), and a light/dark cycle of 14/10 h. All procedures were performed under Biosafety level II conditions. Mice were randomly distributed into two experimental groups: non-infected (n = 5) and infected (n = 5) groups. Mice were infected by intraperitoneal injection of 100 µL of MHV-A59 (6000 PFU) (ATCC VR-764) diluted in sterile PBS. Five days after intraperitoneal infection, mice were weighed and euthanized by cervical dislocation to dissect the liver and lung for RT-qPCR analyses. The experimental protocols were approved by the institutional *Comisión de Ética en el Uso de Animales* (protocol #008-16) and were performed according to national law #18.611 and relevant international laboratory animal welfare guidelines and regulations. 

### 2.9. RNA Isolation, c-DNA Synthesis, and Quantitative Real-Time PCR (RT-qPCR)

Total RNA was isolated with Quick-Zol (Kalium technologies, Buenos Aires, Argentina) according to the manufacturer’s protocol. cDNAs were synthesized with TransScript One-Step gDNA Removal and cDNA Synthesis SuperMix (Transgen Biotec, Beijin, China) using random primers. Taq DNA Polymerase (Invitrogen, Waltham, MA, USA) was used for real-time PCR amplification in a QuantStudio 3 Real-Time PCR System (Thermo Fisher Scientific, Waltham, MA, USA), using the primers listed in Table 1. *PPIA* and *Gapdh* were used as the internal reference genes. Data were analyzed using the method of 2^−∆∆CT^ [37]. Quantification of viral load was performed as previously described [38], using the following primers for MHV detection. 

### 2.10. Statistical Analyses

Wilcoxon rank sum test was performed to determine statistical differences between categorical groups. Age was categorized according to the WHO guidelines (“World Health Organization 2020—Novel Coronavirus (2019-nCoV) Situation Report-1,” n.d.): <30 years old, every 10 years between 30–70 years old, and ≥70 years old. Two-sided, increasing, and decreasing Jonckheere–Terpstra trend tests (with 500 permutations) were used to determine gene expression trends among age groups. To standardize the color scale when plotting heatmaps of multiple gene expression values stratified by age, all values were normalized to the youngest age group (<30). To study pairwise correlations between continuous variables, Spearman’s rank correlation coefficient was calculated. Multilinear regression analyses were performed to determine the correlation between the expression of two genes and viral load. To estimate the regression coefficients of the different models, we used a multivariable regression, including gene expression, viral load, and age as covariates. Statistical significance was set as *p* ≤ 0.05. 

## 3. Results

### 3.1. Analysis of IFN-γ Pathway in COVID-19-Positive and -Negative Patients

To study the relevance of the IFNs production and downstream signaling during SARS-CoV-2 infection, we used the publicly available RNA sequencing (RNA-seq) dataset GSE152075, which consists of transcriptomic data from nasopharyngeal swabs from 430 SARS-CoV-2-positive and 54 SARS-CoV-2-negative patients (Figure 1A). Patient demographics are available in Appendix A. To assess whether COVID-19 infection was associated with differential activation of the IFN pathways, we performed a gene set enrichment analysis (GSEA) among COVID-19-positive vs. COVID-19-negative patients. Regarding the IFN-γ pathway, results showed that both regulation and response to IFN-γ were differentially activated at the RNA level in COVID-19-positive patients (Figure 1B). As expected, IFN-I and IFN-III pathways were significantly activated at the RNA level in positive patients (Appendix A). Results showed that the degree of pathway activation was associated with viral load in COVID-19-positive patients, while non-COVID-19 patients presented the lowest GSVA scores (Figure 1C and Appendix A). We also evaluated whether activation of the IFN-γ pathway was associated with the immune response by estimation of immune cell type proportions from bulk gene expression data using the CIBERSORT tool [33]. We performed an unsupervised clustering analysis, which grouped patients in 5 different clusters, and we observed a group of patients with a higher proportion of M1 macrophages, enriched in patients with higher viral load (Figure 1D). No other immune population showed a clear clustering correlated with viral load in this analysis.

Next, we performed extensive bibliographic research to obtain a list of the IFN-γ-associated genes and classified them into two categories: (1) genes belonging to the canonical pathway and (2) genes belonging to the non-canonical pathways [39] (Figure 2A and Appendix A, respectively). First, we evaluated the expression of the six IFN-γ-canonical pathway genes (Figure 2B). Results showed a differential expression of three out of six genes analyzed when comparing COVID-19-positive vs. -negative patients: *IFNGR1* expression was decreased (*p* = 0.028) (Figure 2B(ii)), while *JAK2* and *STAT1* expressions were increased (*p* ≤ 0.001 for both genes) (Figure 2B(v,vi), respectively). No significant differences were observed for the rest of the genes. Next, we analyzed the expression of 26 IFN-γ-related genes included in the non-canonical pathways related to the immune response, cell proliferation, and cell cycle regulation (Figure 2A and Appendix A). A decrease in the expression of the following genes was also observed in COVID-19-positive patients: *RAPGEF1* (*p* = 0.0091), *MAP2K1* (*p* = 0.038), *CEBPB* (*p* = 3.3 × 10^−6^), *STAT6* (*p* = 0.041), *JUN* (*p* = 1.4 × 10^−8^), *PRKACA* (*p* = 0.021), and *AKT1* (*p* = 1.3 × 10^−6^) (Appendix A). Additionally, an increase in the expression of *MAP2K6, CBL* and *RUNX3* was observed in COVID-19-positive patients (*p* = 0.047; *p* = 0.00093; and *p* = 0.0044, respectively) (Appendix A).

Next, we evaluated whether gene expression might be associated with patients’ sex and age as COVID-19-associated risk factors. Although no significant differences were found when comparing gene expression based on patients’ sex (Figure 2C(i) and Appendix A), the expressions for the following genes belonging to the IFN-γ canonical pathway, decreased with age in COVID-19-positive patients: *IFNG* (*p*-trend_decreasing_ = 0.002), *IFNGR2* (*p*-trend_decreasing_ = 0.04), *JAK1* (*p*-trend_decreasing_ = 0.002), *JAK2* (*p*-trend_decreasing_ = 0.002) and *STAT1 (p*-trend _decreasing_ = 0.002) (Figure 2C(ii), right panel). Additionally, the expressions of *STAT3* (*p*-trend_decreasing_ = 0.01), *RUNX3* (*p*-trend_decreasing_ = 0.002), *RAPGEF1* (*p*-trend_decreasing_ = 0.004), *RAP1A* (*p*-trend_decreasing_ = 0.004), *RAC1* (*p*-trend_decreasing_ = 0.002), *PTK2B* (*p*-trend_decreasing_ = 0.01), *MAPK1* (*p*-trend_decreasing_ = 0.002), *MAP3K1* (*p*-trend_decreasing_ = 0.004), *MAP2K6* (*p*-trend_decreasing_ = 0.02), *CPKL* (*p*-trend_decreasing_ = 0.004), *CBL* (*p*-trend_decreasing_ = 0.002), *AKT1* (*p*-trend_decreasing_ = 0.008), and *CREBBP* (*p*-trend_decreasing_ = 0.01) decreased with age (Appendix A). On the contrary, there was no association between gene expression and age in non-COVID-19 patients (Figure 2C(ii), top panel and Appendix A). Of note, both *STAT1* and *JAK2*, whose expression is significantly increased during SARS-CoV-2 infection, decrease their expression in older patients (Figure 2C(ii)). 

### 3.2. Gene Correlation Analysis of Dysregulated IFN-γ-Associated Genes in COVID-19-Positive and -Negative Patients

Further, in order to increase the interpretability of the data and to acquire a better understanding of the results obtained in the differential gene expression study, we performed a pairwise correlation analysis between the 13 genes that showed a differential expression between COVID-19-positive and -negative patients (Figure 3A). Patients were segregated into three groups: all patients (group 1), non-COVID-19 patients (group 2), and COVID-19 patients (group 3). Results showed significant (*p* < 0.05) and intermediate (0.33 < |r| < 0.66) correlations among several IFN-γ-associated genes for COVID-19-positive patients: *MAP2K6/JAK2* (r = 0.569, *p* < 0.001); *CEBPB/JAK2* (r = 0.466, *p* < 0.001); *CEBPB/CEBL* (r = 0.457, *p* < 0.001); *CEBPB/RUNX3* (r = 0.440, *p* < 0.001); *JUN/JAK2* (r = 0.480, *p* < 0.001); *JUN/STAT1* (r = 0.410, *p* < 0.001); *JUN/CBL* (r = 0.385, *p* < 0.001); *JUN/RAPGEF1* (r = 0.495, *p* < 0.001); *JUN/MAP2K6* (r = 0.346, *p* < 0.001); *PRKACA/JAK2* (r = 0.419, *p* < 0.001); *PRKACA/CBL* (r = 0.518, *p* < 0.001); *AKT1/JAK2* (r = 0.441, *p* < 0.001); *AKT1/STAT1* (r = 0.510, *p* < 0.001) and *AKT1/CBL* (r = 0.525, *p* < 0.001), whereas correlations for the same pairs of genes in non-COVID-19 patients were non-significant (Figure 3A). Interestingly, although *JAK2* and *STAT1*, two of the main genes involved in the IFN-γ canonical pathway, presented a positive and significant correlation for non-COVID19 (r = 0.390, *p* < 0.01), this correlation was significantly increased in COVID-19-positive patients (r = 0.82, *p* < 0.001) patients (Figure 3A). Further, this correlation was associated with viral load in univariate analysis and multivariable analysis, including age as covariable (Figure 3B,C). These results suggest that *STAT1* and *JAK2* might be key responders to SARS-CoV-2 infection.

### 3.3. SARS-CoV-2 Viral Load Association with IFN-γ-Related Genes and ISGs

Taking into account our previous results, we next analyzed the association between the expressions of *JAK2, STAT1*, and four key ISGs relevant to the antiviral response: *MX1*, *MX2*, *ISG15*, and *OAS1* (Figure 4A). The expression of these ISGs was higher in COVID-19-positive compared with COVID-19-negative patients (Figure 4B). Furthermore, results evidenced that COVID-19-positive patients with higher expressions of *JAK2* and *STAT1* showed increased *MX1, MX2*, *ISG15*, and *OAS1* mRNA levels (Figure 4C). Since *MX1*, *MX2*, *ISG15*, and *OAS1* showed a similar expression pattern in COVID-19-positive patients, we next evaluated the association of their combined expressions with viral load. Results showed that higher levels of these 4 ISGs are associated with a higher viral load (Figure 4D). Moreover, the combined expression of *MX1*, *MX2*, *ISG15*, and *OAS1* showed a significant and positive correlation with the combined expression of *JAK2* and *STAT1* (Figure 4E). 

Finally, we assessed the association between the combined expression of *JAK2* + *STAT1* and *MX1* + *MX2* + *ISG15* + *OAS1* with viral load using linear univariable regression analysis. Results evidenced that viral load is associated with an increase in the combined expression of *JAK2* + *STAT1* (*p* = 0.008) and *MX1* + *MX2* + *ISG15* + *OAS1* (*p* = 0.005) (Figure 4F(i)). Furthermore, when analyzing the individual expression of the selected genes, we observed that viral load can explain variations in *JAK2* (*p* = 0.048), *STAT1* (*p* = 0.007), *MX1* (*p* < 0.001), *ISG15* (*p* = 0.002) and *OAS1* (*p* < 0.001). Non-significant differences were observed for *MX2* association with viral load (*p* = 0.35) (Figure 4F(ii)). In conclusion, these results suggest that viral load influenced *JAK2*, *STAT1*, *MX1, ISG15*, and *OAS1* expression; however, its effect is prominent in *MX1* levels. 

### 3.4. Targets of IFN-γ Pathway in Response to Viral Infection

To further assess the alterations in *MX1*, *MX2*, *ISG15*, and *OAS1* in response to viral infections and validate our results, we performed in vitro assays in A549 cells using Poly (I:C), a synthetic analog of viral double-stranded RNA, to mimic a viral infection (Figure 5A(i)). We found that *MX1, MX2, ISG15*, and *OAS1* expressions were significantly increased in A549 cells treated with Poly (I:C) 10 µM during 24 h (*p* < 0.05 for *MX1, MX2*, and *OAS1*; and *p* < 0.001 for *ISG15*) (Figure 5A(ii)). Additionally, we evaluated the 4 ISGs response against single-stranded RNA viruses, such as Influenza A virus (IAV), Respiratory syncytial virus (RSV), Middle East respiratory Syndrome (MERS-CoV), and SARS-CoV-1 (Appendix A). As expected, *MX1*, *MX2*, *ISG15* and *OAS1* were up-regulated in human airway epithelial cells (hAEC) infected with IAV (*p* < 0.001 for all genes), RSV (*p* < 0.001 for all genes), MERS-CoV (*p* < 0.001 for *MX1*, *ISG15* and *OAS1*; and *p* < 0.01 for *MX2*) and SARS-CoV-1 (*p* < 0.001 for *MX1* and *ISG15*; and *p* < 0.01 for *MX2*) compared with mock treatment; confirming *MX1*, *MX2*, *ISG15*, and *OAS1* overexpression during different viral infections (Appendix A). We next assessed *MX1*, *MX2*, *ISG15*, and *OAS1* expression levels in response to SARS-CoV-2 infection (Figure 5B,C). We used the GSE147507 dataset, which contains RNA-seq expression data from lung cell lines (A549, Calu-3, and NHBE), as well as tracheal biopsies of SARS-CoV-2-infected ferrets. Results showed that *MX1*, *MX2* and *OAS1* were over-expressed in SARS-CoV-2-infected A549 cells (*p* < 0.01 for *MX1*, *ISG15*, and *OAS1*; and *p* < 0.05 for *MX2*), and Calu-3 cells (*p* < 0.05 for *MX1*, *ISG15*, and *OAS1*; and *p* < 0.01 for *MX2*) compared with mock infection (Figure 5B). In NHBE cells, only *MX1*, *MX2*, and *OAS1* were significantly over-expressed in SARS-CoV-2-infected cells compared with mock infection (*p* < 0.05 for *MX1* and *OAS1*; and *p* < 0.01 for *MX2*) (Figure 5B). Further, SARS-CoV-2 infection significantly induced these ISGs in ferret tracheal biopsy samples collected on day 3 after infection compared with mock-treated animals (*p* < 0.01 for *MX1*; and *p* < 0.05 for *MX2, ISG15*, and *OAS1*) (Figure 5C). Altogether, these results confirm that the expression of *MX1*, *MX2*, *ISG15*, and *OAS1* was triggered by SARS-CoV-2 infection.

Finally, we validated our results by studying the expression of *Mx1*, *Mx2*, *Isg15*, and *Oas1* in a murine model of Coronavirus infection using the mouse hepatitis virus (MHV), a type 2 family RNA Coronavirus similar to SARS-CoV-2. We selected this model as an alternative pre-clinical model for the study of Coronavirus infection. BALB/cJ mice were infected with 6000 PFU of MHV-A59 by intraperitoneal injection, and five days after infection, liver and lung were dissected for RT-qPCR analyses (Figure 6A). First, we confirmed that the infection was successful by qPCR analysis of the viral load. Results showed that viral infection was detectable in the liver and lungs, the main organs affected by MHV (Figure 6B(i,ii)). We next evaluated the expression of *Mx1*, *Mx2*, *Isg15*, and *Oas1* in non-infected vs. infected mice and found that all four genes were increased in the liver (*p* < 0.01 for *Mx1* and *Isg15*; and *p* < 0.05 for *Mx2* and *Oas1*) and lungs (*p* < 0.01 for *Mx1*, *Mx2* and *Isg15*; and *p* < 0.05 for *Oas1*) of infected mice (Figure 6C). Our results validate the relevance of these four ISGs against Coronavirus infection. 

## 4. Discussion

The aim of the present work was to assess the implication of the IFN-γ pathway in response to SARS-CoV-2 infection. Results from extensive bioinformatics analyses in a case-control study from COVID-19-positive (n = 430) and -negative (n = 54) patients identified a signature of 13 dysregulated IFN-γ-associated genes, including *STAT1* and *JAK2*, both key mediators of the IFN-γ signaling. *STAT1/JAK2* positively correlated with *MX1*, *MX2*, *ISG15*, and *OAS1*, four well-known genes regulated by IFN [14]. Further, higher levels of these ISGs combined showed a significant association with the viral load. We validated the relevance of *MX1*, *MX2*, *ISG15*, and *OAS1* in the context of viral infections in cell cultures in vitro and in a pre-clinical model of Coronavirus infection [40]. 

IFNs are essential key players against viral infections through the induction of ISGs, that work in synergy to inhibit the replication and spread of the virus [41,42]. However, IFNs show differential associations with clinical markers of poor prognosis and COVID-19 severity [43]. For example, IFN-β1, a type of IFN-I, is associated with an increased neutrophil to lymphocyte ratio, a marker of late severe disease; while IFN-γ, a type of IFN-II, is strongly associated with C-reactive protein and other immune markers of poor prognosis [44]. In this context, previous studies in SARS-CoV infections reported that IFN-β might be a valid candidate to treat COVID-19, and a recent clinical trial demonstrated that IFN-β1a reduced the duration of patients’ hospital stay and/or ameliorated their clinical status [44]. 

In this work, we evidenced the alteration of IFN-γ-associated genes during SARS-CoV-2 infection. Our results rendered 13 dysregulated genes when comparing COVID-19-positive vs. -negative patients, suggesting an IFN-γ-related gene expression signature characteristic of COVID-19 disease. Comparatively, Galbraith et al. performed a multi-omics investigation of systemic IFN signaling in hospitalized SARS-CoV-2-infected patients with varying levels of IFNs [43]. Of note, it has been reported that higher expression levels of IFN-γ in the upper airway determine the pathogenesis of COVID-19 [45], as it correlates with higher levels of ACE2, thus increasing susceptibility to infection [45]. It has also been reported that IFN-γ-driven inflammatory responses contribute to SARS-CoV-2 replication [46]. Further, IFN-γ contributes to COVID-19 complications, such as cytokine storm, tissue damage, and inflammation [47], since an exacerbated immune response may also cause damage to the host. Previous reports evidenced that this cytokine is over-expressed in patients that died of COVID-19 when comparing its expression in patients who survived, highlighting it as a mortality risk factor [48]. Regarding age as a critical epidemiological risk factor, we found that the expression levels of genes belonging to the IFN-γ canonical pathway decreased with age in COVID-19-positive patients. This observation suggests that there might be a correlation between immune antiviral response impairment and age-associated changes in the expression of IFN-γ after SARS-CoV-2 infection. However, we cannot rule out that this might be a random observation instead of a causative association. Interestingly, Bastard et al. reported that neutralizing autoantibodies against IFN-I increase with age [49]. Of note, extensive bibliography suggested the link between the presence of IFN-γ autoantibodies and a lower immune response to several opportunistic pathogenic agents is related to an increased risk of infections [50,51,52,53,54]. In light of this, we hypothesized that IFN-γ autoantibodies could diminish COVID-19 response. Accordingly, other studies reported that in critically ill COVID-19 patients, IFN-γ administration was followed by viral clearance and clinical improvement [55,56]. Moreover, no information is available about the expression of IFN-γ downstream genes in this cohort of patients. Additionally, it has also been suggested that enhancing STAT1’s activity might be an effective COVID-19 treatment [57], thus bypassing the adverse effects of IFN-based therapeutics. On the other hand, JAK inhibitors have been tested in hospitalized severe COVID-19 patients showing an improvement in treatment results alone or combined with standards of care (i.e., corticosteroids) [58,59,60,61].

The insights into the dysregulation of IFN pathways in COVID-19 patients provide a biological basis for new therapies. Further, targeting IFN downstream genes might be advantageous considering the pleiotropic activities of human IFNs [62]. Notably, in the present work, we demonstrated that the expressions of the ISGs *MX1*, *MX2*, *ISG15*, and *OAS1* were upregulated in COVID-19-positive vs. -negative patients.

*MX1* encodes a guanosine triphosphate (GTP)-metabolizing protein that inhibits the replication of RNA and DNA viruses in the nucleus, while *MX2* is a GTPase that has a reported potent antiviral activity in the cytoplasm [63]. In a recent work from our group, we highlighted the relevance of *MX1* as an antiviral SARS-CoV-2 host effector [29]. Here, we validated *MX1* and *MX2* relevance in vitro and in a valuable murine model of Coronavirus infection. Not much information is available concerning *MX1* and *MX2* during SARS-CoV-2 infection; however, in melanoma cell lines, it was reported that persistent IFN-γ stimulation increases the expression and genome occupancy of STAT1 and induces the expression of ISGs, such as *IFIT1* and *MX1* [64]. A study of red blood cells’ methylome demonstrated a hypomethylation of *MX1* in samples from COVID-19-hospitalized patients, highlighting the relevance of the INF-γ pathway modulation during SARS-CoV-2 infection [65]. 

*ISG15* encodes a ubiquitin-like protein that interacts with viral proteins during infection, resulting in their loss of function [66,67]. Further, free extracellular ISG15 acts as an amplifier of inflammation, exacerbating the production of cytokines by macrophages [68]. Regarding the mechanisms that the SARS-CoV-2 employs in order to evade the antiviral immune response, it has been reported that the viral papain-like protease (PLpro) acts as a deISGylation enzyme that cleaves ISG15 from its substrates, contributing to the evasion of the host immune system [69]. 

OAS1 participates in the innate cellular antiviral response. This enzyme synthesizes oligomers that activate RNAse L, which degrades cellular and viral RNA and impairs viral replication [70]. Danziger et al. found that OAS1 catalytic activity is required for antiviral response against SARS-CoV-2 [71]. Moreover, a membrane-associated OAS1 form could act as a ds-RNA sensor in infection sites. Further, *OAS* was identified as a COVID-19 risk locus in association studies suggesting that the splice-site variant at this locus may reduce the enzymatic activity of OAS-1, influencing COVID-19 outcomes [72,73,74,75,76]. Recent evolutionary studies about the human immune response against viral infections point out that OAS1 plays an important role in SARS-CoV-2 pathogenesis [73,77]. Furthermore, Hurguin et al. reported an over-expression of *OAS1*, *MX1*, and *MX2* after IFN-γ induction of a HeLa-derived cell line [78], confirming that the IFN-γ signaling promotes the expression of our ISGs signature. 

Taking into account the current public health emergency of international concern, it is essential to increase the development of alternative treatment options against COVID-19. Although much attention has been placed on virus host cell receptors, little has been explored on COVID-19 therapies targeting effector pathways and antiviral proteins [79]. The main therapeutic avenues to halt respiratory virus infection consist of targeting the virus directly or targeting the host system. Even though the first strategy is highly efficient, it is limited by the sensitivity/resistance of the virus variants [80,81]. Thus, deciphering the host response to viral infection appears critical. This study aimed to explore the relevance of the IFN-γ signaling pathway during SARS-CoV-2 infection with the long-term goal of delineating new therapeutic targets for COVID-19. However, all the knowledge about the mechanisms that are triggered in the host cell as a consequence of the infection could also be applied to treat other viral infections (i.e., Dengue or Zika).

## Figures and Tables

**Figure 1 viruses-14-02180-f001:**
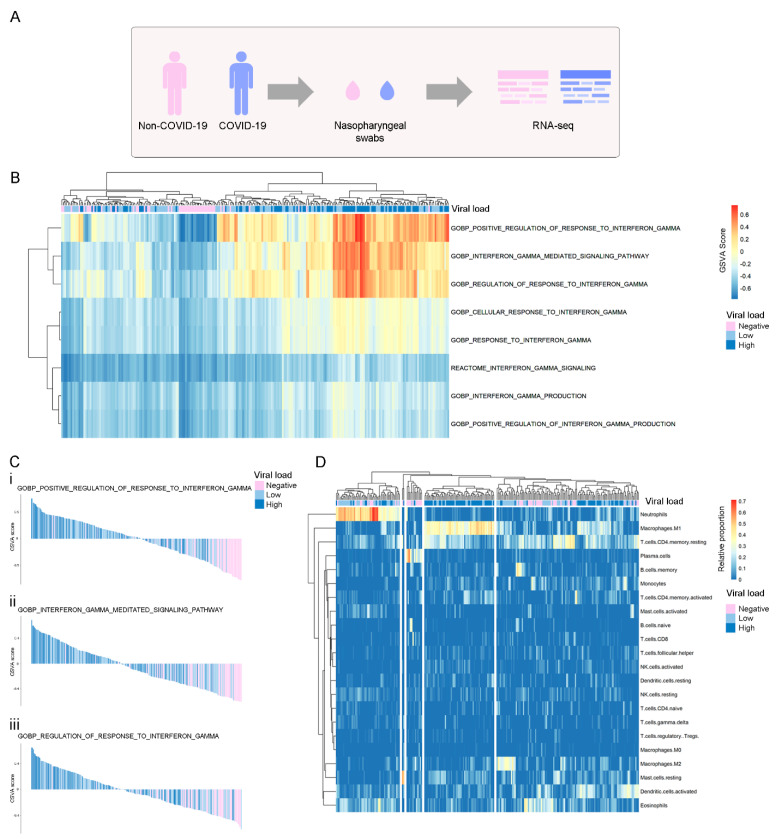
Global assessment at the transcriptional level of pathways and immune cell types related to IFN-γ production, signaling, and regulation of response in COVID-19-positive (light blue and blue) and -negative (pink) patients. (**A**) Experimental design of the GSE152075 dataset, composed of transcriptome data from nasopharyngeal swabs collected from 430 COVID-19 and 54 non-COVID-19 patients. (**B**) Non-supervised clustering of patients according to their GSVA score in each IFN-γ geneset. Higher GSVA scores indicate higher activity of the geneset at the RNA level. Each column is labeled according to the COVID-19 viral load of each patient (negative: pink, low: light blue, high: blue). (**C**) Waterfall plots of selected genesets that were highly activated in COVID-19 patients vs. non-COVID-19 patients. Patients are ordered from the highest to the lowest GSVA score in each geneset. (**D**) Unsupervised clustering of non-COVID-19 and COVID-19 patients according to the relative proportions of immune cell types estimated by CIBERSORT (LM22 signature) in RNA-seq data. Each column is labeled according to the COVID-19 viral load of each patient. Viral load is represented as a color scale and was categorized as Negative (pink), Low (first quartile; light blue), or High (fourth quartile; blue). COVID-19 patients with intermediate viral load were excluded from the analysis.

**Figure 2 viruses-14-02180-f002:**
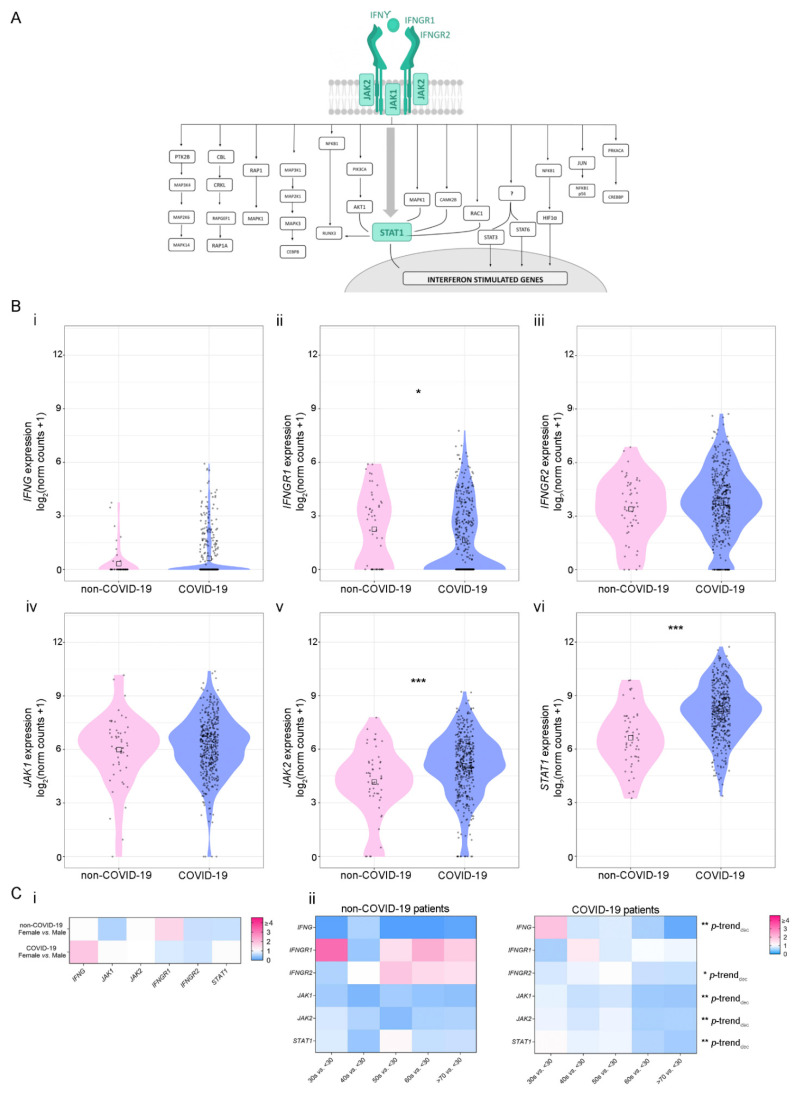
Expression of genes belonging to the canonical IFN-γ pathway in non-COVID-19 and COVID-19 patients. (**A**) Schematic representation of IFN-γ signaling pathway associated genes. The canonical IFN-γ pathway is represented in green. (**B**) Gene expression analysis (log2 (norm counts +1)) for (i) *IFNG*, (ii) *IFNGR1*, (iii) *IFNGR2*, (iv) *JAK1*, (v) *JAK2*, and (vi) *STAT1* in COVID-19 (purple) vs. non-COVID-19 (pink) patients from the GSE152075 dataset, assessed by RNA-seq. *p*-values correspond to Wilcoxon rank-sum test. Black squares represent the median. (**C**) Heatmaps depicting the fold change (high = pink; low = blue) for gene expression of genes belonging to the canonical IFN-γ pathway considering sex (i) and age groups 30 s, 40 s, 50 s, 60 s, and 70 s vs. <30 (ii) in non-COVID-19 (left panel) and COVID-19 (right panel) patients from the GSE152075 dataset, assessed by RNA-seq. *p*-values correspond to decreasing Jonckheere–Terpstra trend test. Statistical significance * *p* < 0.05; ** *p* < 0.01; *** *p* < 0.001.

**Figure 3 viruses-14-02180-f003:**
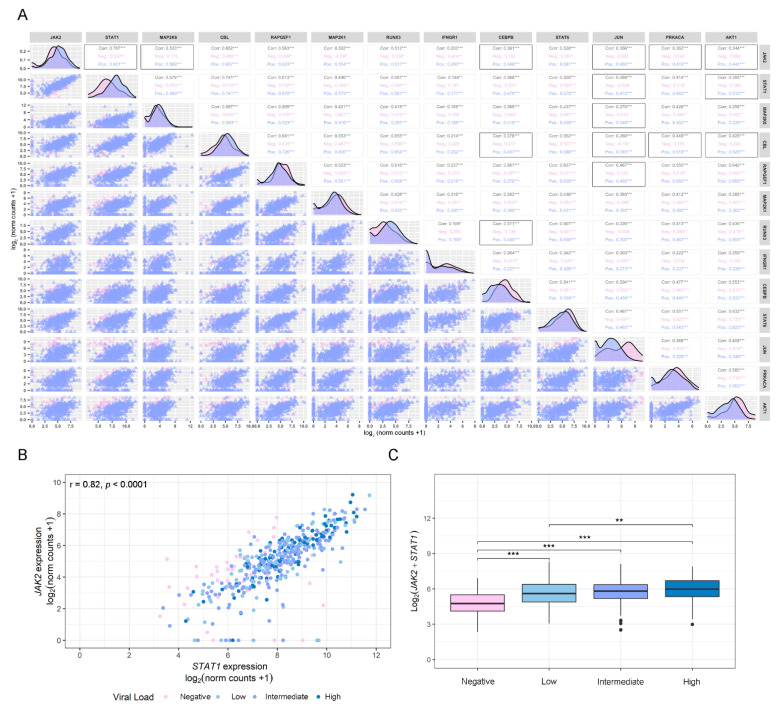
Correlation analysis. (**A**) Pairwise Spearman correlation matrix analysis between all genes of interest using the GSE152075 dataset. The upper half displays the Spearman coefficients (r) considering all patients (Corr.; black), non-COVID-19 patients (Neg.; pink), or COVID-19 patients (Pos.; purple). Black boxes highlight pairs of genes that have significant correlation only in COVID-19-positive patients, except for *JAK2*/*STAT1*, which was the pair with the highest coefficient in COVID-19-positive patients. The lower half displays the scatterplots. (**B**) Dot plot representing pairwise Spearman correlation for *JAK2* and *STAT1*, considering viral load in COVID-19-positive patients from the GSE152075 dataset. Viral load is represented as a color scale and was categorized as Negative (pink), Low (first quartile; light blue), Intermediate (second and third quartile; purple), or High (fourth quartile; blue), and was considered as an independent variable expressed as cycle threshold (Ct) by RT-qPCR for the N1 viral gene at time of diagnosis. The interpretation for viral load is the lower the Ct, the higher the viral load. (**C**) Box plot representing the combined expression of *JAK2* + *STAT1* and their association with the viral load. Viral load was categorized as Negative (pink), Low (first quartile; light blue), Intermediate (second and third quartile; purple), or High (fourth quartile; blue). *p*-values correspond to Wilcoxon rank-sum test. Statistical significance * *p* < 0.05; ** *p* < 0.01; *** *p* < 0.001.

**Figure 4 viruses-14-02180-f004:**
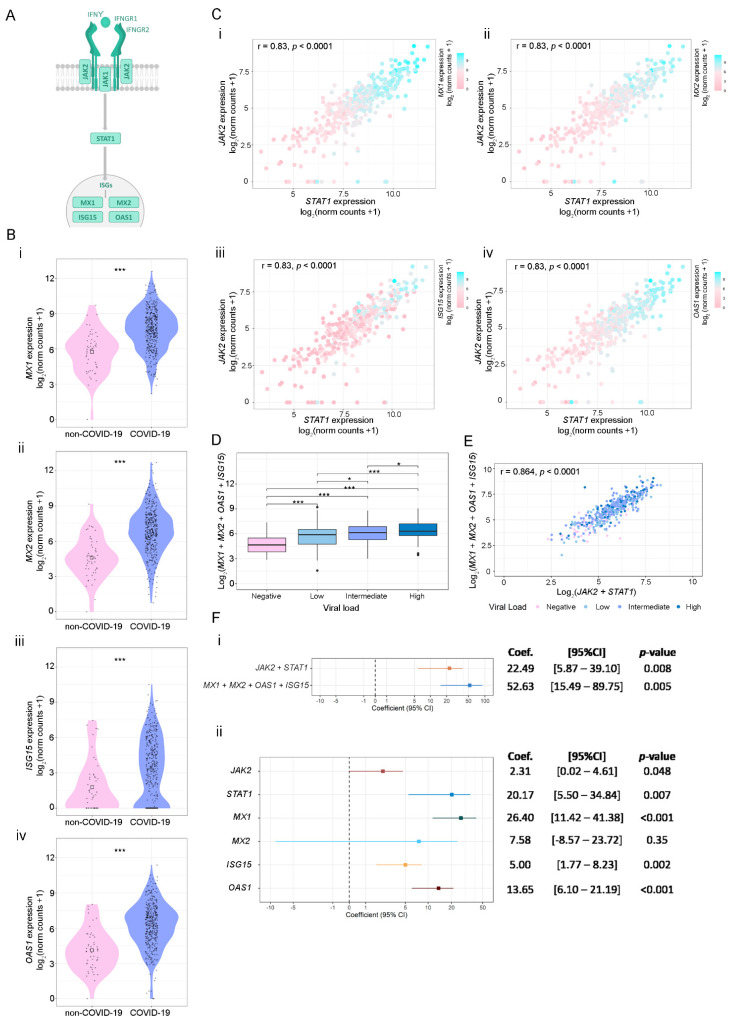
Association between *STAT1*, *JAK1*, and ISG’s expressions using the GSE152075 dataset. (**A**) Schematic representation of the canonical IFN-γ signaling pathway. (**B**) Gene expression analysis (log2 (norm counts +1)) for *MX1* (i), *MX2* (ii), *ISG15* (iii), and *OAS1* (iv) in COVID-19 (purple) vs. non-COVID-19 (pink) patients from the GSE152075 dataset, assessed by RNA-seq. *p*-values correspond to Wilcoxon rank-sum test. (**C**) Dot plots representing the pairwise Spearman correlation between *STAT1* and *JAK1*, considering the expressions of *MX1* (ii), *MX2* (ii), *ISG15* (iii), and *OAS1* (iv) in COVID-19-positive patients. The independent variable is plotted on the *x* axis, and the dependent variable is plotted on the *y* axis. *MX1*, *MX2*, *ISG15* and *OAS1* expressions are represented as a color scale (purple = high; pink = low). (**D**) Box plot representing the combined expression of *MX1* + *MX2* + *ISG15* + *OAS1* and their association with the viral load. Viral load was categorized as Negative (pink), Low (first quartile; light blue), Intermediate (second and third quartile; purple), or High (fourth quartile; blue). *p*-values correspond to Wilcoxon rank-sum test. (**E**) Dot plot representing pairwise Spearman correlation between *JAK2* + *STAT1* and *MX1* + *MX2* + *ISG15* + *OAS1* combined expressions, considering viral load in COVID-19-positive patients from the GSE152075 dataset. Viral load is represented as a color scale and was categorized as Negative (pink), Low (first quartile; light blue), Intermediate (second and third quartile; purple), or High (fourth quartile; blue). (**F**) Forest plots representing the coefficient of the viral load as predictor variable in regression analyses (considering age and gender as covariables). (i) Model considering as response variable: combined expression of *JAK2* + *STAT1* or *MX1* + *MX2* + *ISG15* + *OAS1*; and (ii) model considering response variable: individual expressions of *JAK2*, *STAT1*, *MX1*, *MX2*, *ISG15* and *OAS1*. Coef.: Coefficient. CI: Confidence interval. Statistical significance * *p* < 0.05; ** *p* < 0.01; *** *p* < 0.001.

**Figure 5 viruses-14-02180-f005:**
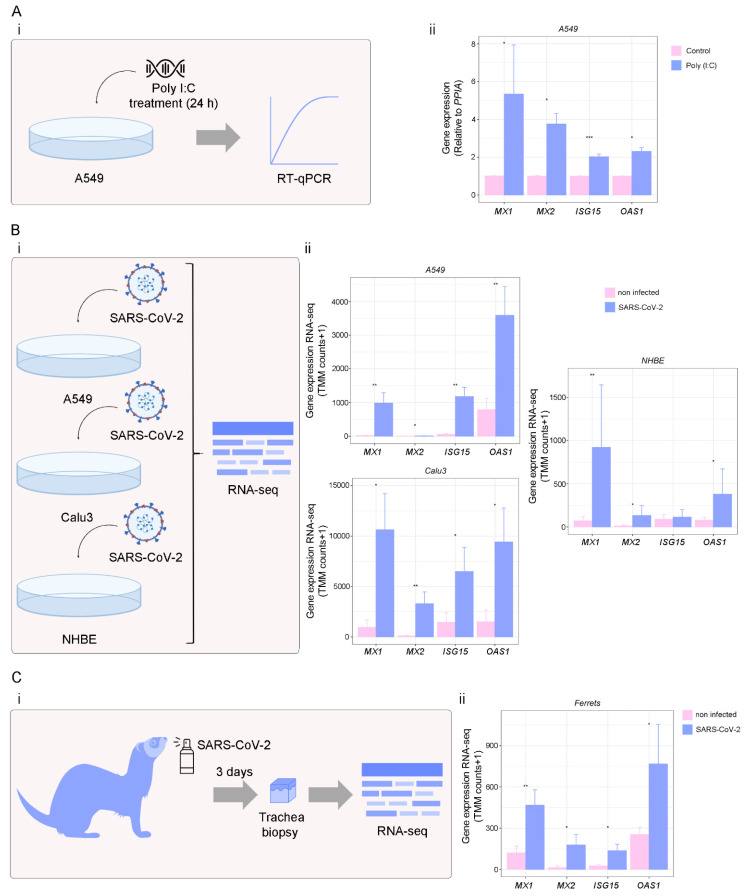
Analysis of ISGs expression in viral infections. (**A**) A549 cells were treated with Poly (I:C) for 24 h, and RNA was extracted for RT-qPCR analysis. (i) Schematic representation of the experimental design. (ii) *MX1*, *MX2*, *ISG15*, and *OAS1* expressions assessed by RT-qPCR in A549 cells treated (purple) or not (pink) with Poly (I:C) (10 µg/ml; 24 h). Values were relativized using *PPIA* as a reference gene and normalized to the control. (**B**) *MX1*, *MX2*, *ISG15*, and *OAS1* expressions (norm counts +1) in SARS-CoV-2-treated cells. (i) Schematic representation of the experimental design. (ii) *MX1*, *MX2*, *ISG15*, and *OAS1* expressions in SARS-CoV-2-infected A549 (n = 6), Calu3 (n = 6), and NHBE (n = 10) cell lines (purple) (MOIs: 0.2, 2, and 2, respectively for 48 h) compared with mock (pink), assessed by RNA-seq, using the GSE147507 dataset. (**C**) *MX1*, *MX2*, *ISG15*, and *OAS1* expressions (norm counts +1) in SARS-CoV-2-treated ferrets. (i) Schematic representation of the experimental design. (ii) *MX1*, *MX2*, *ISG15*, and *OAS1* expressions in SARS-CoV-2-infected (5 × 10^4^ PFU) (purple) vs. mock-treated (pink) ferrets, assessed by RNA-seq in tracheal biopsy samples (n = 7), using the GSE147507 dataset. Samples were collected on day 3 after SARS-CoV-2 infection. Student’s *t*-test was performed to determine statistical differences. Statistical significance * *p* < 0.05; ** *p* < 0.01; *** *p* < 0.001.

**Figure 6 viruses-14-02180-f006:**
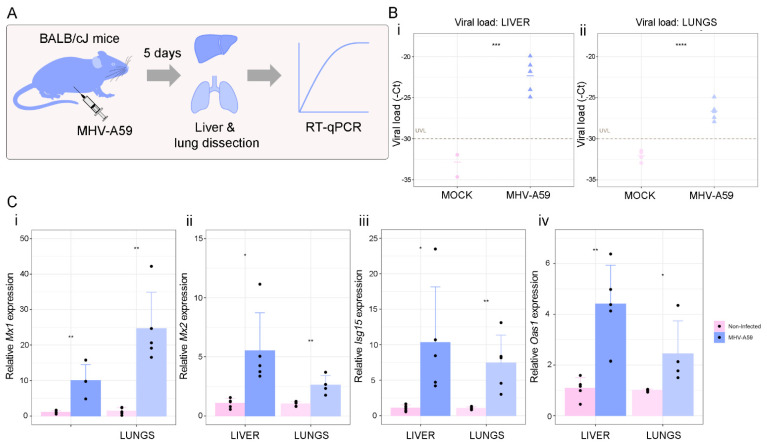
Analysis of ISGs expression in a pre-clinical model of Coronavirus infection. (**A**) Schematic representation of the experimental design. BALB/cJ female mice (n = 10) were used for the in vivo experiments. Mice were randomly distributed into two experimental groups: non-infected (n = 5) and infected (n = 5) groups. Mice were infected with 6000 PFU of MHV-A59 by intraperitoneal injection of 100 µL of the virus diluted in sterile PBS. Five days after infection, mice were weighed and euthanized by cervical dislocation to dissect the liver and lung for RT-qPCR analyses. (**B**) Viral load was assessed by RT-qPCR in livers (i) and lungs (ii) of BALB/cJ mice infected (purple) or not (pink) with MHV-A59. Viral load is expressed as-cycle threshold (-Ct) by RT-qPCR for MHV. UVL: undetectable viral load. (**C**) *Mx1* (i), *Mx2* (ii), *Isg15* (iii), and *Oas1* (iv) expressions assessed by RT-qPCR in livers and lungs of BALB/cJ mice infected (purple) or not (pink) with MHV-A59. Values were relativized using *Gapdh* as a reference gene and normalized to the control. Student’s *t*-test was performed to determine statistical differences. Statistical significance * *p* < 0.05; ** *p* < 0.01; *** *p* < 0.001; **** *p* < 0.0001.

**Table 1 viruses-14-02180-t001:** List of primers used in RT-qPCR.

Gene	Forward (5′-3′)	Reverse (5′-3′)	T ann (°C) *
*MX1*	AGGACCATCGGAATCTTGAC	TCAGGTGGAACACGAGGTTC	60
*MX2*	GGCAGAGGCAACCAAGAAAGA	AACGGGAGCGATTTTTGGA	60
*ISG15*	GTCCTGCTGGTGGTGGACAAA	GTCCTGCTGGTGGTGGACAAA	61
*OAS1*	GGGATTTCGGACGGTCTTGG	TCTCCACCACCCAAGTTTCC	60
*PPIA*	GGTATAAAAGGGGCGGGAGG	CTGCAAACAGCTCAAAGGAGAC	60
*Mx1*	TGCCAGGACCAGGTTTACAAG	CCCCTTTTGAGGAAACTGAGA	58
*Mx2*	CCTATTCACCAGGCTCCGAA	TCTCGTCCACGGTACTGCTT	58
*Isg15*	TGAGAGCAAGCAGCCAGAAG	CCCCCAGCATCTTCACCTTT	57
*Oas1*	ACTTCCTGAACTGTCGCCC	ACTCGACTCCCATACTCCCAG	61
*Gapdh*	TGCCAAGGCTGTGGGCAAGG	CGAAGGTGGAAGAGTGGG	60
MHV	GGAACTTCTCGTTGGGCATTATACT	ACCACAAGATTATCATTTTCACAACATA	60

* T ann = Annealing temperature.

## Data Availability

Not applicable.

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
