# Peer review of "Pin-Pointing the Key Hubs in the IFN-γ Pathway Responding to SARS-CoV-2 Infection"

_viruses, 2022, doi:10.3390/v14102180_

Round 1
Reviewer 1 Report
The authors seek to determine if genes associated with the IFN-g pathway in response to SARS-CoV-2 infection may be potential targets for COVID-19 treatment. Their primary assessments were bioinformatics analyses of public RNA-seq and microarray data from primary lung cell lines infected with SARS-CoV-2 or MERS-CoV as well as COVID-19 and MERS patients. They conclude that 13 IFN-g associated genes are regulated in COVID-19 patients compared to negative people. They also validated this conclusion using the mouse model infected with hepatitis virus. Overall, this manuscript answered an interesting question in the field that what can be targeted to COVID-19 treatment, however, big major concerns are it is not clear what is critical new findings from this manuscript and the authors used data set from one paper that analyzed COVID-19 patients.
The IFN-g related genes that are induced by SARS-CoV-2 infection have been identified in COVID-19 patients who got infected by ancestral virus or variants, and primary lung cells treated by IFN-g or SARS-CoV-2 infection (PMID: 32413319, 33138195, 33841438, 34308298, 34685525, 35402859, 35465056 and so on). Also, there are papers that addressed the relationship between mutations on OAS genes and severity of COVID-19 disease (PMID: 33307546, 34581622, 35835913 and so on). Many groups have generated RNA-seq and scRNA-seq data from COVID-19 patients with different level of severity that can be connected viral load. However, this manuscript missed those papers. Although 68 papers are cited in this manuscript, many are from 2020 or before the pandemic.
1) Although the authors presented a heatmap of viral load, it is difficult to identify groups of negative, low, and high viral load in Figure 1B, 1C, D, 3A.
2) It’s hard to read name and numbers of X- and Y-axis in all figures.
3) There are five heatmaps in Figure 1D. However, it’s not clear how the groups are divided.
4) Authors said “IFNGR1 expression was decreased” in Figure 2B ii (line 277). However, unlike to B-v and vi, this violin plot is not shown clear difference between non-COVID-19 and COVID-19 groups. Please add median in the plot.
5) Authors said “26 IFN-g-related genes included in the non-canonical pathways related to three categories” in Figure 2A and Table S2 (line 280-282). Please present genes in these three categories instead of gene list.
6) Figure S3 was not described in the result part.
7) Authors described some genes are decreased with age in COVID-19 positive patients (line 302-314). However, there is no interpretation about why the genes are related to age of COVID-19 patient. Please add potential pathway or any immune response related to age and disease.
8) Authors used a word “non-COVID-19 patients” in the manuscript. Were they asymptomatic patients? It’s not clear in the manuscript.
9) Since some figures have low resolution, it’s difficult to read. For example, Figure 3A and 4A.
10) To confirm the conclusion, authors used mice infected by hepatitis virus as Corona virus infection model. However, there are many papers used COVID-19 mouse models infected by SARS-CoV-2 (PMID: 32854108, 33166988, 35045565, 32732280 and so on). Evidence about whether both viruses show similar infection mechanism, induce the same immune response, and so on is needed.
11) Authors said “STAT1’s activity might be an effective COVID-19 treatment, thus bypassing the adverse effects of IFN-based therapeutics.” in the discussion section (line 514-515). However, several papers have examined the effect of JAK inhibitors in COVID-19 treatment, and they have been used as COVID-19 treatment, especially patients with cytokine storm. It is needed to be added latest references and information in the discussion section.
Author Response
Response to reviewer #1:
The authors seek to determine if genes associated with the IFN-g pathway in response to SARS-CoV-2 infection may be potential targets for COVID-19 treatment. Their primary assessments were bioinformatics analyses of public RNA-seq and microarray data from primary lung cell lines infected with SARS-CoV-2 or MERS-CoV as well as COVID-19 and MERS patients. They conclude that 13 IFN-g associated genes are regulated in COVID-19 patients compared to negative people. They also validated this conclusion using the mouse model infected with hepatitis virus. Overall, this manuscript answered an interesting question in the field that what can be targeted to COVID-19 treatment, however, big major concerns are it is not clear what is critical new findings from this manuscript and the authors used data set from one paper that analyzed COVID-19 patients.
The IFN-g related genes that are induced by SARS-CoV-2 infection have been identified in COVID-19 patients who got infected by ancestral virus or variants, and primary lung cells treated by IFN-g or SARS-CoV-2 infection (PMID: 32413319, 33138195, 33841438, 34308298, 34685525, 35402859, 35465056 and so on). Also, there are papers that addressed the relationship between mutations on OAS genes and severity of COVID-19 disease (PMID: 33307546, 34581622, 35835913 and so on). Many groups have generated RNA-seq and scRNA-seq data from COVID-19 patients with different level of severity that can be connected viral load. However, this manuscript missed those papers. Although 68 papers are cited in this manuscript, many are from 2020 or before the pandemic.
We thank the reviewer for this comment. We acknowledge the limitations regarding the use of only one dataset from COVID-19 patients. We will consider this suggestion for our future analyses. Additionally, as suggested, in this new version of the manuscript we included new and updated bibliography throughout the discussion section: Lee et al., 2013; Liew et al., 2019; Valour et al., 2016; Shih et al., 2021; Danziger et al., 2022; Banday et al., 2022, Wickenhagen et al., 2021; Pairo-Castineira et al., 2021; Huffman et al., 2022; Kerner and Quintana Murci, 2022; ZHou et al., 2021.
1- Although the authors presented a heatmap of viral load, it is difficult to identify groups of negative, low, and high viral load in Figure 1B, 1C, D, 3A.
We thank the reviewer for his/her comment. We would like to clarify this issue: negative, low and high viral load groups are represented by pink, light blue and blue colors, respectively, as indicated in the heatmap references and above the heat map.
The heatmap in Figure 1B shows the GSVA Score for the different IFN-γ genesets; while Figure 1D heatmap shows the relative proportion of different immune cell types, estimated by CIBERSORT (Newman et al., 2019). In both cases, unsupervised clustering was performed to analyze how these different variables (e.g, positive regulation of response to IFN-γ, IFN-γ mediated signaling pathway, etc) group patients with different viral loads. Results showed that most patients that were classified as “non-COVID-19” because of their negative result of viral load present different GSVA scores (lower, as shown in Figure 1B, 1C). Moreover, in Figure 1D we observe the segregation of a group of patients with higher viral load when performing this analysis, as having a higher proportion of M1 macrophages. Although this heatmap shows viral load, the main aim of this plot is to show how the other categories (e.g, positive regulation of response to IFN-γ, IFN-γ mediated signaling pathway, etc) segregate patients according to their viral load. Viral load levels can be identified at the top of the heatmaps, in the “Viral load” line. On the other hand, Figure 1C is a waterfall plot showing differences in viral load (negative, low or high) according to genesets activation levels. Lastly, Figure 3A is a correlation analysis between dysregulated IFN-γ associated genes in COVID-19 positive and negative patients. In this Figure there are no viral load groups. Pink represents negative COVID-19 patients and purple corresponds to positive COVID-19 patients, regardless of their viral load levels.
In order to clarify this concern, we have amended the text and now reads:
Figure 1. Global assessment at the transcriptional level of pathways and immune cell types related to IFN-γ production, signaling and regulation of response in COVID-19 positive (light blue and blue) and negative (pink) patients. (A) Experimental design of the GSE152075 dataset, composed of transcriptome data from nasopharyngeal swabs collected from 430 COVID-19 and 54 non-COVID-19 patients. (B) Non-supervised clustering of patients according to their GSVA score in each IFN-γ geneset. Higher GSVA scores indicate higher activity of the geneset at the RNA level. Each column is labeled according to the COVID-19 viral load of each patient (negative: pink, low: light blue, high: blue). (C) Waterfall plots of selected genesets that were highly activated in COVID-19 patients vs. non-COVID-19 patients. Patients are ordered from the highest to the lowest GSVA score in each geneset. (D) Unsupervised clustering of non-COVID-19 and COVID-19 patients according to the relative proportions of immune cell types estimated by CIBERSORT (LM22 signature) in RNA-seq data. Each column is labeled according to the COVID-19 viral load of each patient. Viral load is represented as a color scale and was categorized as Negative (pink), Low (first quartile; light blue), or High (fourth quartile; blue). COVID-19 patients with intermediate viral load were excluded from the analysis (lines 261-275).
Figure 3. Correlation analysis. (A) Pairwise Spearman correlation matrix analysis between all genes of interest using the GSE152075 dataset. The upper half displays the Spearman coefficients (r) considering all patients (Corr; black), non-COVID-19 patients (Negative; pink), or COVID-19 patients (Positive; purple). Black boxes highlight pairs of genes that have significant correlation only in COVID-19 positive patients, except for JAK2/STAT1, that was the pair with the highest coefficient in COVID-19 positive patients. The lower half displays the scatterplots (lines 350-355).
2- It’s hard to read name and numbers of X- and Y-axis in all Figures.
We apologize for the inconvenience. We have increased the font sizes in all the graphs.
3- There are five heatmaps in Figure 1D. However, it’s not clear how the groups are divided.
We thank the reviewer for his/her comment. We would like to clarify this issue: in Figure 1D there is only one heatmap representing the segregation of patients with different viral loads (negative: pink, low: light blue or high: blue) according to the relative proportion of different immune populations inferred by CIBERSORT (Newman et al., 2019). This heatmap is the result of an unsupervised clustering analysis, which grouped patients in 5 different clusters. These clusters recapitulate differences in the immune populations. This analysis allowed us to observe that there is a subpopulation of positive COVID-19 patients, mainly classified as “high viral load” which present a higher relative abundance of macrophages M1, as depicted in cluster 4.
In order to clarify this concern, we have amended the text (lines 255-259) and now reads: “We performed an unsupervised clustering analysis, which grouped patients in 5 different clusters, and we observed a group of patients with a higher proportion of M1 macrophages, enriched in patients with higher viral load (Figure 1D). No other immune population showed a clear clustering correlated with viral load in this analysis.”
4- Authors said “IFNGR1 expression was decreased” in Figure 2B ii (line 277). However, unlike to B-v and vi, this violin plot is not shown clear difference between non-COVID-19 and COVID-19 groups. Please add median in the plot.
We thank the reviewer for his/her comment. The median is already included in the Figure, as expressed in Figure 2 legend (lines 297-300): “Gene expression analysis (log2 (norm counts +1)) for (i) IFNG, (ii) IFNGR1, (iii) IFNGR2, (iv) JAK1, (v) JAK2, and (vi) STAT1 in COVID-19 (purple) vs. non-COVID-19 (pink) patients from the GSE152075 dataset, assessed by RNA-seq. p-values correspond to Wilcoxon rank-sum test. Black squares represent the median.” Black squares in the middle of the violin plot represent the median of expression of the different analyzed genes. Of note, several black dots are overlapping on “zero”, thus explaining the diminished expression of IFNGR1.
5- Authors said “26 IFN-g-related genes included in the non-canonical pathways related to three categories” in Figure 2A and Table S2 (line 280-282). Please present genes in these three categories instead of gene list.
We thank the reviewer for this observation. In the revised version of the manuscript, we have now incorporated these three categories in the Figure S3 of the Supplementary information.
6- Figure S3 was not described in the result part.
We apologize for this omission. We have amended the text. We also want to clarify that the old Figure S3 is now Figure S4 because in this new version we have incorporated Figure S3 with the three categories of IFN-γ genes belonging to the non-canonical pathway. Results section now reads:
A decrease in the expression of the following genes was also observed in COVID-19 positive patients: RAPGEF1 (p = 0.0091), MAP2K1 (p = 0.038), CEBPB (p = 3.3e-06), STAT6 (p = 0.041), JUN (p = 1.4e-08), PRKACA (p = 0.021), and AKT1 (p = 1.3e-06) (Supplementary Figure S4A). Additionally, an increase in the expression of MAP2K6, CBL and RUNX3 was observed in COVID-19 positive patients (p = 0.047; p = 0.00093; and p = 0.0044, respectively) (Supplementary Figure S4A) (lines 287-293).
Although no significant differences were found when comparing gene expression based on patients’ sex (Figure 2C i & Supplementary Figure S4B), the expressions for the following genes belonging to the IFN-γ canonical pathway, decreased with age in COVID-19 positive patients: IFNG (p-trenddecreasing = 0.002), IFNGR2 (p-trend decreasing = 0.04), JAK1 (p-trend decreasing = 0.002), JAK2 (p-trend decreasing = 0.002) and STAT1 (p-trend decreasing = 0.002) (Figure 2C ii, right panel). Additionally, the expressions of STAT3 (p-trend decreasing = 0.01), RUNX3 (p-trend decreasing = 0.002), RAPGEF1 (p-trend decreasing = 0.004), RAP1A (p-trend decreasing = 0.004), RAC1 (p-trend decreasing = 0.002), PTK2B (p-trend decreasing = 0.01), MAPK1 (p-trend decreasing = 0.002), MAP3K1 (p-trend decreasing = 0.004), MAP2K6 (p-trend decreasing = 0.02), CPKL (p-trend decreasing = 0.004), CBL (p-trend decreasing = 0.002), AKT1 (p-trend decreasing = 0.008), and CREBBP (p-trend decreasing = 0.01) decreased with age (Supplementary Figure S4C ii). On the contrary, there was no association between gene expression and age in non-COVID-19 patients (Figure 2C ii, top panel & Supplementary Figure S4C i) (lines 307-321).
7- Authors described some genes are decreased with age in COVID-19 positive patients (line 302-314). However, there is no interpretation about why the genes are related to age of COVID-19 patient. Please add potential pathway or any immune response related to age and disease.
We thank the reviewer for his/her comment. We found that several genes belonging to the IFN-γ canonical pathway decreased with age in COVID-19 positive patients. In the new version of the manuscript, we have incorporated a plausible interpretation of this result. The fact that older people who get infected with SARS-CoV-2 present worse symptoms is widely known. Our results show that IFN-γ pathway associated genes gradually decrease with age in COVID-19 positive patients. However, this is not a common feature of age-related changes in gene expression, as non-COVID-19 patients do not present the same trend. These findings led us to hypothesize that the changes in age-associated severity might be connected to an impaired immune antiviral response, generated by these gene expression changes in the elderly.
Discussion section now reads (lines 516-522): “Regarding age as a critical epidemiological risk factor, we found that the expression levels of genes belonging to the IFN-γ canonical pathway decreased with age in COVID-19 positive patients. This observation suggests that there might be a correlation between immune antiviral response impairment and age-associated changes in the expression of IFN-γ pathway after SARS-CoV-2 infection. However, we cannot rule out that this might be a random observation instead of a causative association. ”
8- Authors used a word “non-COVID-19 patients” in the manuscript. Were they asymptomatic patients? It’s not clear in the manuscript.
We thank the reviewer for his/her concern. Throughout the manuscript we refer to “non-COVID-19” and “COVID-19” patients since when the diagnosis was performed (by RT qPCR), positive patients were included in the “COVID-19” group while negative patients were referred as “non-COVID-19” group (GSE152075). In consequence, either asymptomatic or symptomatic COVID patients were included in the “COVID-19” group.
For further clarification we have amended the methodology section (lines 126-127) and now reads: “RNA-seq data from 430 SARS-CoV-2 positive (COVID-19) and 54 negative patients (non-COVID-19) diagnosed by RT-qPCR.”
9- Since some Figures have low resolution, it’s difficult to read. For example, Figure 3A and 4A.
We thank the reviewer for this observation. We apologize for this inconvenience. We have submitted high quality images (.TIFF) for all the Figures. However, it appears as if during the PDF conversion process, the quality was lost. We will make sure if the manuscript is approved for publication to re-check the quality of the Figures.
10- To confirm the conclusion, authors used mice infected by hepatitis virus as Corona virus infection model. However, there are many papers used COVID-19 mouse models infected by SARS-CoV-2 (PMID: 32854108, 33166988, 35045565, 32732280 and so on). Evidence about whether both viruses show similar infection mechanism, induce the same immune response, and so on is needed.
We thank the reviewer for his/her concerns. It is well accepted that the mouse hepatitis virus (MHV) is an excellent model for studying the pathogenesis including tropism and virulence, as well as immune response to Coronaviruses and was used as a model for SARS-CoV (Körner et al., 2020). Thus, the MHV model could offer mechanistic insights into SARS-CoV-2 and new promising avenues for treatment of COVID-19. MHV and SARS-CoV-2 share various similarities. Regarding the infection mechanism, the Spike protein determines host spectrum and tropism of both Coronaviruses, mediating the fusion between the viral and host cell membrane. MHV, as well as SARS-CoV-2, is highly contagious, strongly immunomodulating and it alters the interferon responsiveness of infected mice (Körner et al., 2020). Further, it was demonstrated that the MHV strain employed in the present work (MHV-A59) replicates in the lung and induces severe lung injuries (Yang et al., 2014) and the recent work of Arevalo et al. supports and validates this mouse experimental model to study Coronavirus infection (Arevalo et al., 2021). Additionally, please note that in our experiments we used mice which are the appropriate host for MHV.
11- Authors said “STAT1’s activity might be an effective COVID-19 treatment, thus bypassing the adverse effects of IFN-based therapeutics.” in the discussion section (line 514-515). However, several papers have examined the effect of JAK inhibitors in COVID-19 treatment, and they have been used as COVID-19 treatment, especially patients with cytokine storm. It is needed to be added latest references and information in the discussion section.
We thank the reviewer for the valuable suggestion. Multiple studies have evaluated the efficiency of treating hospitalized COVID-19 patients with JAK inhibitors. Some of them have shown improvements in patient's condition, though others have not proven to be exceedingly more beneficial than standard treatments (Guimarães et al., 2021; Wolfe et al., 2022; Levy et al., 2022; Hasan et al., 2021; Marconi et al., 2021). Nevertheless, the use of JAK inhibitors in severe COVID-19 patients has been approved by the WHO (https://www.covid19treatmentguidelines.nih.gov). Interestingly JAK inhibitors effectiveness is given not only by its anti-inflammatory effects but also by its direct antiviral activity. We have amended the discussion regarding JAK inhibitors in COVID-19 treatment and included the corresponding references.
The Discussion section now reads (lines 532-535): “On the other hand, JAK inhibitors have been tested in hospitalized severe COVID-19 patients showing an improvement in treatment results alone or combined with standards of care (i.e. corticosteroids) (Guimarães et al., 2021; Wolfe et al., 2022; Marconi et al., 2021; RECOVERY Collaborative Group, 2022; Levy et al., 2022; Hasan et al., 2021; https://www.covid19treatmentguidelines.nih.gov).”

Reviewer 2 Report
In this manuscript, Toro et al. have analyzed the IFN-g pathway following SARS-CoV-2 infection. Their analysis started from data available in public databases, which are however supported by wet experiments.
Although it is difficult to establish whether this analysis has a significant relevance in identifying a therapeutic approach for SARS-CoV-2 infections, the experiments shown in the manuscript are carried out in a technically rational manner and with adequate controls.
In my opinion, the manuscript can be published in its current version.
Author Response
Response to reviewer #2:
In this manuscript, Toro et al. have analyzed the IFN-g pathway following SARS-CoV-2 infection. Their analysis started from data available in public databases, which are however supported by wet experiments.
Although it is difficult to establish whether this analysis has a significant relevance in identifying a therapeutic approach for SARS-CoV-2 infections, the experiments shown in the manuscript are carried out in a technically rational manner and with adequate controls.
In my opinion, the manuscript can be published in its current version.
We thank the reviewer for his/her comments. Since extensive bibliography focuses on type I and III IFN pathways and its impaired response during COVID-19 pathogenesis, and little is known about type II IFN in this disease, in the present work we aimed to explore the relevance of the IFN-γ signaling pathway during SARS-CoV-2 infection with the long-term goal of delineating new therapeutic targets for COVID-19. We found that IFN-γ signaling induces the expression of MX1, MX2, ISG15 and OAS1 as SARS-CoV-2 associated antiviral genes in human COVID-19 patients. We also state that these antiviral genes respond to diverse viral infections including SARS-CoV-2 infection in vitro and validate our findings in a preclinical murine model of Coronavirus infection. Thus, deciphering a crucial feature of the host antiviral response pointing out potential targets to halt SARS-CoV-2 infection.

Reviewer 3 Report
In this manuscript, Toro et al., compared several ISGs expression from COVID-19 donors. They analyzed large-scale data from both non-COVID-19 and COVID-19 donors and showed that higher levels of ISGs was associated with higher viral loads and STAT1/JAK2 levels. They also examined these hypotheses in animal models. All experiments were well performed and data/scheme is clear. However, I have a few comment on this manuscript.
minor
1. Regarding the COVID-19 patient samples, do the authors know if the donors showed mild or severe symptoms?
2. Do the authors know/speculate with which variants the donors were infected?
3. Figure 2A: the words in the figure are too small to read. Can the authors fix it?
4. Figure 6B: The authors showed viral load as Ct value. The authors should show virus loads as “relative viral loads” or “copy/ml”.
5. Figure 6B: please add in the Figure legend what “UVL” means.
6. Please fix these in the Figure legend.
Figure 2 legend: There are no **P<0.01 in the Figure. Please delete it.
Figure 3 legend: There are no *P<0.05 in the Figure. Please delete it.
Figure 6 legend: ****P is missing in the legend. Please add it.
Figure 5 legend: The authors described “ns: not significant”. But there are no “ns” in the Figure. Please add this in the Figure or delete this.
Author Response
Response to reviewer #3:
In this manuscript, Toro et al., compared several ISGs expression from COVID-19 donors. They analyzed large-scale data from both non-COVID-19 and COVID-19 donors and showed that higher levels of ISGs was associated with higher viral loads and STAT1/JAK2 levels. They also examined these hypotheses in animal models. All experiments were well performed and data/scheme is clear. However, I have a few comment on this manuscript.
Minor:
1- Regarding the COVID-19 patient samples, do the authors know if the donors showed mild or severe symptoms?
We thank the reviewer for his/her comments. We examined the dataset GSE152075 which have information about infection status, viral load, age, and sex from 430 individuals infected with SARS-CoV-2 and 54 negative controls. Unfortunately, this dataset does not contain information regarding the level of severity and its association with viral load.
2- Do the authors know/speculate with which variants the donors were infected?
We thank the reviewer for his/her comments. The dataset GSE152075 was generated from samples obtained in 2020. For this reason, we speculate that COVID-19 positive individuals were infected with SARS-CoV-2 B.1 variant.
3- Figure 2A: the words in the Figure are too small to read. Can the authors fix it?
We apologize for this inconvenience. We have increased the font sizes in all the Figures.
4- Figure 6B: The authors showed viral load as Ct value. The authors should show virus loads as “relative viral loads” or “copy/ml”.
We thank the reviewer for her/his suggestion. The PCR cycle threshold (Ct) is a semi-quantitative value inversely related to the initial amount of genetic material in the sample and therefore it can be used as an estimate of the viral load (Rico-Caballero et al., 2022). We consider that “genome copies per ml” could be a better way to show viral load but this information is not available in the dataset GSE152075. To express viral load in copies per ml a standard solution is necessary to perform a calibration curve using viral genome standard (Puahch et al., 2022) which is not included in the dataset that we analyzed. However, Ct value is used as a surrogate variable for viral load in the original research paper that published this dataset, and in several other works (i.e., Levine-Tiefenbrun et al., 2021 and 2022). Considering the dataset limitations, we believe that Ct it is also an appropriate form to express viral load.
5- Figure 6B: please add in the Figure legend what “UVL” means.
We apologize for this omission. We have amended the legend of Figure 6 (lines 473-476) which now reads: “(B) Viral load was assessed by RT-qPCR in livers (i) and lungs (ii) of BALB/cJ mice infected (purple) or not (pink) with MHV-A59. Viral load is expressed as - cycle threshold (-Ct) by RT-qPCR for MHV. UVL: undetectable viral load. “
6.- Please fix these in the Figure legend:
Figure 2 legend: There are no **P<0.01 in the Figure. Please delete it.
In Figure 2C ii there are ** in JAK1, JAK2 and STAT1. For this reason, we conserve **P<0.01 in the legend of the Figure (line 305).
Figure 3 legend: There are no *P<0.05 in the Figure. Please delete it.
In Figure 3A (correlation matrix) there are some *. For this reason, we conserve *P<0.05 in the legend of the Figure (lines 361-362).
Figure 6 legend: ****P is missing in the legend. Please add it.
We apologize for this omission. We have added ****P<0.0001 in the legend of Figure 6 (line 480).
Figure 5 legend: The authors described “ns: not significant”. But there are no “ns” in the Figure. Please add this in the Figure or delete this.
We apologize for this mistake. We have removed “ns: not significant” from the legend of Figure 5 (line 452).

Reviewer 4 Report
The manuscript of Toro et al is a fascinating manuscript that highlights the importance of the IFN gamma pathway in SARS CoV2 infection the manuscript has an important amount of data, and it has been well organized. However, I would like to point out several issues that may improve the manuscript
1. - According to the literature, and part of the authors have published a computational analysis, A549 is not the best model of SARS CoV2 infection; there are other options in which even the modulation of IFN alpha and omega can be quantified. Could the authors clarify?
2.-In figure 1 part D it is important to address the issue of similar responses independent of the viral load, part B of the figure.
3.-Figure 2 part C the IFN in part 1 and ii, age correspondence, one may infer that the amount of young women volunteers predominates in the analysis. However, table S1 states that there are different ages. The authors were not convincing in their argument.
4.-Figure 3 part A is very low quality and difficult to read, but SAT1 and JUN seem to be the most relevant. Part C of the figure is the real interesting data, which should be analyzed correction with age and gender. I would suggest that part A of the figure should be in a supplemental file
5.-The groups of Drs Cananova and Bastard have documented the importance of autoantibodies anti-IFN gamma among others. How will autoantibodies affect the cell response? and the anti-viral response?
The quality of the figures should be enhanced.
Author Response
Response to reviewer #4:
The manuscript of Toro et al is a fascinating manuscript that highlights the importance of the IFN gamma pathway in SARS CoV2 infection the manuscript has an important amount of data, and it has been well organized. However, I would like to point out several issues that may improve the manuscript.
1- According to the literature, and part of the authors have published a computational analysis, A549 is not the best model of SARS CoV2 infection; there are other options in which even the modulation of IFN alpha and omega can be quantified. Could the authors clarify?
We thank the reviewer for this observation. The RNA-seq from the dataset GSE147507 was obtained in A549 cells transduced with a vector expressing human ACE2 (hACE2 A549). We have clarified this in the manuscript in the Materials and Methods section that now reads (lines 121-125): RNA-seq data from: (i) human cell lines derived from primary bronchial/tracheal epithelial cells (NHBE), lung carcinoma (hACE2 A549) and lung adenocarcinoma (Calu-3) infected with SARS-CoV-2 (MOIs: 2, 0.2 and 2, respectively) or mock-PBS; and (ii) tracheal samples from ferrets intranasally infected with 5×104 PFU of SARS-CoV-2 or mock-PBS.
Consequently, it is a suitable model for studying SARS-CoV-2 infection. On the other hand, our in vitro experiments were performed in wild type A549 cells. However, we employed this cell line to mimic a viral infection by transfection with Poly(I:C), a synthetic analog of viral double-stranded RNA which agonizes TLR3. In effect, for our purposes we consider this human lung cell line as a suitable experimental model. Additionally, it is important to highlight that even in this cell line, we could observe an effect in accordance with the results in other cell lines (Calu-3 and NHBE) and in the murine model of Coronavirus infection.
2- In Figure 1 part D it is important to address the issue of similar responses independent of the viral load, part B of the Figure.
We thank the reviewer for this observation. In the new version of the manuscript, we have improved the analysis of Figure 1D. The 3.1 Results section now reads (lines 255-259): “We performed an unsupervised clustering analysis, which grouped patients in 5 different clusters, and we observed a group of patients with a higher proportion of M1 macrophages, enriched in patients with higher viral load (Figure 1D). No other immune population showed a clear clustering correlated with viral load in this analysis.”
3- Figure 2-part C the IFN in part 1 and ii, age correspondence, one may infer that the amount of young women volunteers predominates in the analysis. However, table S1 states that there are different ages. The authors were not convincing in their argument.
We thank the reviewer for his/her comment. We would like to clarify our findings: Table S1 shows that 56% of non-COVID19 patients are female. Thus, they do not predominate in the analysis. We did not find any significant differences in IFN-γ gene expression when comparing males and females; therefore, we did not consider gender as a confounder when analyzing the changes in gene expression associated with age groups.
We did not find differences in the gene expression of the different analyzed genes in non-COVID-19 patients, while we found significant trends (p-trends) for 5 of the 6 analyzed genes in COVID-19 positive patients when exploring age-associated differences.
For further clarification we have modified the text in Materials and Methods section that now reads: “To standardize the color scale when plotting heatmaps of multiple genes expression values stratified by age, all values were normalized to the youngest age group (< 30s). (lines 225-227)”
4- Figure 3 part A is very low quality and difficult to read, but SAT1 and JUN seem to be the most relevant. Part C of the Figure is the real interesting data, which should be analyzed correction with age and gender. I would suggest that part A of the Figure should be in a supplemental file.
We thank the reviewer for this observation. We apologize for this inconvenience. We have submitted high quality images (.TIFF, 2480 pixels x 3508 pixels) for all the Figures. However, it appears as if during the PDF conversion process, the quality was lost. We will make sure if the manuscript is approved to re-check the quality of the Figures.
We agree with the reviewers on the importance of controlling for covariables. We performed independent analysis to control the combined expression of STAT1 and JAK2, for the age and gender of the patients. There is no association of the gender with the combined expression of STAT1 and JAK2. Regarding the age of the patients, univariable and multivariable analysis including age as covariable showed a significant association of the viral load and the combined expression of STAT1 and JAK2.
We have incorporated these clarifications in the results, the text now reads (lines 343-344): “Further, this correlation was associated with viral load in univariate analysis and multivariable analysis including age as covariable”
5- The groups of Drs Cananova and Bastard have documented the importance of autoantibodies anti-IFN gamma among others. How will autoantibodies affect the cell response? and the anti-viral response?
We thank the reviewer for this valuable comment. We have incorporated in the discussion section the relevance of autoantibodies anti-IFN-γ.
The Discussion section now reads (lines 522-527): “Interestingly, Bastard et al. reported that neutralizing autoantibodies against IFN-I increase with age. Of note, extensive bibliography suggested the link between the presence of IFN-γ autoantibodies and a lower immune response to several opportunistic pathogenic agents are related with an increased risk of infections. In light of this, we hypothesized that IFN-γ autoantibodies could diminish COVID-19 response.”
6- The quality of the Figures should be enhanced.
We thank the reviewer for this observation. We apologize for this inconvenience. We have submitted high quality images (.TIFF) for all the Figures. However, it appears as if during the PDF conversion process, the quality was lost. We will make sure if the manuscript is approved to re-check the quality of the Figures.
